# The Link between Sleep Insufficiency and Self-Injury among In-School Adolescents: Findings from a Cross-Sectional Survey of Multi-Type Schools in Huangpu District of Shanghai, China

**DOI:** 10.3390/ijerph192315595

**Published:** 2022-11-24

**Authors:** Shan Zhang, Chunyan Yu

**Affiliations:** 1Department of Comprehensive Prevention and Emergency Management, Huangpu District Center for Disease Control and Prevention, Shanghai 200023, China; 2NHC Key Laboratory of Reproduction Regulation (Shanghai Institute for Biomedical and Pharmaceutical Technologies), Fudan University, Shanghai 200237, China

**Keywords:** self-injury, insufficient sleep, middle school, coping, mental health

## Abstract

Both insufficient sleep and self-injury are rising public health issues among middle school students. Understanding their relationship may guide the intervention and policy making to help youths gain a healthy life. Thus, we analysed the data collected from the Shanghai Students Health Risk Behavior Surveillance (2015) in the Huangpu District. Self-injury was self-reported and categorized into ever or never. Sleep duration was classified as sufficient and insufficient according to the Health China 2030 Plan and the National Sleep Foundation’s updated sleep duration recommendations. Crude OR and adjusted OR of sleep duration and covariates were estimated for self-injury using the logistic regression models with standard error clustered on school types. Results showed that 8.42% of the participants had conducted self-injury, with girls more than boys and ordinary school students more than key school students. After full adjustment, sleep insufficiency increased the odds of conducting self-injury by approximately two folds (AOR = 2.08, 95%CI = 1.40–3.07). The odds of self-injury were higher among students studying at ordinary schools (AOR = 3.58, 95%CI = 1.25–10.27) or vocational schools (AOR = 2.00, 95%CI = 1.77–2.26), with comparison to those at key schools. Interventions seeking to solve insufficient sleep need to be multifaceted, with consideration of changing the school environment and multiple social contexts, which create stressful burdens for adolescents’ development.

## 1. Introduction

Self-injury among adolescents is a major public health concern worldwide, with an approximately 10–20% 1-year prevalence rate, which is still rising [1,2]. A school-based study among European adolescents aged 14–17 years old showed that about 14.6% had thought about self-injury, while 5.8% had carried out so during the past year. In China, according to a meta-analysis conducted in 2018, the self-injury rate among middle-school students was 22.37% (95%CI: 18.84–25.70%) [3,4]. Major risk factors include mental disorders, bully, low health literacy, problem behaviors, adverse childhood experiences, physical symptoms, and female gender [5]. In addition, there has been growing evidence that sleep problems are risk factors for self-injury and suicidal behavior, and that this relationship is independent of the psychiatric disorder [6,7].

Sufficient sleep is beneficial not only for energy conservation, neuronal recovery, and brain plasticity, but also for physical growth, and cognitive and psychological development [8]. According to the National Sleep Foundation in the US, the time spent on sleep should be at least 9 h, 8 h, and 7 h for school-age children, teenagers, and young adults, respectively [8]. The Health China 2030 Plan also suggested ensuring at least 10 h of sleep for children in primary school, 9 h for those in junior high school, and 8 h for those in senior high school. However, the sleep duration of many adolescents falls well short. In the US, adolescent self-reported sleep has decreased over the past 20 years [9]. In Asian countries such as Singapore and Korea, sleep duration of five or six hours per night is common [10]. Insufficient sleep duration is detrimental to adolescent development, and poses essential and complicated health risks in the adolescent population [11]. Research has reported that insufficient sleep is negatively associated with academic performance, self-control, depression, juvenile delinquency, substance use, and other risk-taking behaviors towards oneself, such as self-injury and suicidal behaviors [7,12,13].

Behaviors such as suicide and self-injury tend to cluster in a particular group [14]. The social contagion theory, which is based on social learning theory, suggests that such behaviors provide others with role models and increase the likelihood of imitating the behavior [15]. As school constitutes one of the most critical environments influencing the adolescents’ health behaviors and well-being through their established friendships, perceived social norms, and academic expectations, the effect of schools, especially different school types, should be noticed. Specifically, in the current Chinese context, schooling is becoming an essential investment for the family [16]. After the 9-year compulsory schooling, about half of the junior high students who wanted to pursue higher education would go to senior high school, with the “cleverest” ones being preferred by key senior high schools and the ordinary students going to the ordinary senior high school. In contrast, the other half would go to vocational schools, with more focus on skill training to prepare for future work. With tremendous academic pressure, Chinese students who want to perform better are even forced to sacrifice sleep time [17]. To our knowledge, however, few studies on the topic of self-injury in China have taken this into consideration. The present study aims to expand the state of knowledge in the area.

## 2. Participants and Methods

### 2.1. Participants and Data Collection

The current study is conducted in the middle schools located in the Huangpu District, Shanghai in 2015 as a part of Shanghai Students Health Risk Behavior Surveillance (SSHRS), guided by the Shanghai Municipal Center for Disease Control and Prevention (CDC) and operated by Huangpu District CDC. The aim of SSHRS is to obtain surveillance of adolescent health risk behaviors and the possible factors. This study recruited 6th to 12th grade students from ordinary, key, and vocational schools to complete a questionnaire. This study used a two-stage cluster sample design to create a representative sample of middle school students in the Huangpu District. The study protocol for conducting SSHRS was approved by the Institutional Review Board (IRB) in the CDC of the Huangpu District. Parental consent and adolescent assent were collected with the help of school teachers.

### 2.2. Measurements

#### Outcome Variable: Self-Injury

Self-injury in this study was measured based on response to the question “During the past 12 months, have you injured yourself intentionally and how many times?”. Adolescents who answered “0 times” were coded as 0, and those who answered “1 time”, “2–3 times”, “4–5 times”, and “6 times or more” were coded as 1.

### 2.3. Explanatory Variable: Insufficient Sleep

Insufficient sleep was measured as a binary variable. The original question asked respondents “On an average night, how many hours of sleep do you get?” with the following response options “less than 7 h”, “7–8 h”, “8–9 h” and “over 9 h”. Following the recommendations of the Health China 2030 Plan and the National Sleep Foundation’s updated sleep duration recommendations [8], adolescents who were aged less than 14 years and had less than 9 h of sleep (“less than 7 h”, “7–8 h”, “8–9 h”) on an average night; who were aged 14–17 and had less than 8 h of sleep (“less than 7 h”, “7–8 h”); and who were aged over 18 and had less than 7 h of sleep were considered as having an insufficient sleep and were recoded as 1, whereas adolescents who had chosen equivalent more hours of sleep on an average night according to their age were considered as having sufficient sleep and were recoded as 0.

### 2.4. Covariates

To better capture the association between self-injury and sleep insufficiency, we controlled for other potential risk factors for self-injury: self-rated weight, school performance [18], bully victimization [5], mental health disturbances [3,19], substance use (cigarette and alcohol use) [20], and time spent on video games [3,21]. Physical activity was considered a protective factor that could reduce the possibility of self-injury and was measured by asking, “During the past 7 days, on how many days were you physically active for a total of at least 60 min per day?” Respondents who were physically active on five or more days were considered physically active and were coded as 1; otherwise, they were considered physically inactive and were coded as 0.

Self-rated weight was measured by the question “How would you describe your body weight?” and the answers were categorized into three groups: very light and a little light were combined as “light than average”; a little heavy and very heavy were combined as “heavy than average”; and the average group remained the same and was treated as the reference group in the multivariate analysis.

School performance was self-reported and the answers were categorized into three groups as similar as self-rated weight: very good and a little good were combined as “better off”; a little worse and very bad were incorporated into “worse off”; and the average group remained the same and was treated as the reference group in the multivariate analysis.

Bullying victimization was measured based on the response to a set of questions, “During the past 12 months, have you ever been bullied in the following forms? (1) Being teased; (2) Being robed; (3) Being intentionally excluded from group activity or isolated; (4) Being threatened; (5) Being physically beaten, kicked, attacked, or squeezed; (6) Being sexually harassed”. In either circumstance, adolescents could choose: never (counted as 0), occasionally (counted as 1), or often (counted as 1). Then, we calculated them in tertials based on the principal component scores of all five items of the respondents [22]. The top 1/3 were considered as heavily bullied, while the middle and bottom 1/3 were characterized as moderate and none, respectively.

A mental problem was measured using four items: “During past 12 months, have you worried about the safety on the way between your home and school?”; “During past 12 months, have you felt lonely?”; “During the past 12 months, have you felt unhappy because of the pressure from study and school performance”; and “During the past 12 months, have you stopped normal activities because of sadness and despair?” Those who answered yes were coded as 1; those who answered no were coded as 0. The total score was grouped into tertials based on principle component scores such as bully victimization.

Adolescents who smoked a cigarette at least once during the past 30 days before the survey date were coded as 1; otherwise, they were coded as 0 for smoking activity. Adolescents who reported drinking alcohol at least once during the 30 days before the survey date were coded as 1; otherwise, they were coded as 0 for drinking activity. 

We also adjusted for age, sex, ethnicity, and family structure. Age was measured as a continuous variable, whereas sex was coded as a binary variable with the male as the reference category. Ethnicity was dichotomized into Han ethnicity and minorities, while the family structure was divided into two groups: living with both parents (1), and living with only one parent or no parents (0).

### 2.5. Data Analysis and Statistics

Data analytic strategies included the use of descriptive, bivariate, and multivariate techniques. The general distribution of the study variables was first examined using percentages. Next, we examined the bivariate association between insufficient sleep and the study variables using binary logistic regression. We then regressed self-injury on insufficient sleep using binary logistic regression, while controlling for the effects of demographic and other risk factors for self-injury. A multivariate logistic regression model was then fitted with all the variables assessed in bivariate logistic regression in consideration of interactions. Adjusted odds ratios (AOR) are reported together with their 95% confidence intervals (CI). Variables were considered significant if the *p* value was less than 0.05. All the analyses were performed using STATA version 15.1 SE (StataCorp, College Station, TX, USA).

## 3. Results

### 3.1. Sample Characteristics

Table 1 compared the distribution of respondents’ characteristics by sex and by school type, respectively. Of the 1092 participants, 48.90% were male. Students from ordinary schools, key schools, and vocational schools were 22.07%, 37.27%, and 40.66%, respectively. About 7.68% of male students and 9.14% of female students reported having self-injury experiences, while 12.03% were from ordinary schools. A large majority of adolescents (81.87%) reported having insufficient sleep, with girls slightly more than boys (83.87% vs. 79.78%), and students from key schools (91.89%) more than from ordinary schools (76.35%) or from vocational schools (75.68%). 

### 3.2. Bivariate Association between Self-Injury, Sleep Insufficiency, and Study Variables, Respectively

Table 2 shows the bivariate association between self-injury and the study variable. We found younger age, studying in an ordinary school, self-rating of worse-off school performance, and ever smoking and drinking during the past 30 days were associated with higher self-injury possibilities. Bully experiences and mental health problems were also correlated with self-injury. We further explored the effect of each study variable on sleep insufficiency. The results were shown in Table 3. Compared to key schools, there was an approximately 70% drop off in the odds for students in ordinary schools (OR 0.28, 95%CI: 0.18–0.45) and vocational schools (OR 0.27, 95%CI: 0.18–0.42) to have sleep insufficiency. Smoking experiences during the past 30 days were negatively associated with insufficient sleep by an average decreased odd of 52% (OR: 0.48, 95%CI: 0.24–0.94). Interestingly, there was a negative association between bully experiences and sleep insufficiency, whereby adolescents being heavily bullied were 32% less likely to experience sleep insufficiency. Factors positively associated with sleep insufficiency include higher self-rated weight versus averaged weight (OR: 1.55, 95%CI: 1.09–2.20) and moderated mental health problems versus light mental health problems (OR: 1.95, 95%CI: 1.37–2.77).

### 3.3. The Effect of Sleep Insufficiency on Self-Injury

Table 4 depicts the effect of sleep insufficiency and covariates on self-injury, solely and together using bivariate and multivariate logistic regression with a clustered standard error on school type, as school type correlated with almost every predictor variable. Bivariate results showed that older age, living with both parents, and average and better school performance were negatively associated with self-injury. At the same time, sleep insufficiency, being as a minority, having smoked or drank during the past 30 days, having severe or moderate bully victimization experiences, and having severe or moderate mental health problems were positively correlated with self-injury. 

In the multivariate logistic model, considering the interaction between sleep insufficiency and school type, sleep insufficiency contributes about a two-fold increase in the odds of conducting self-injury (AOR = 2.08, 95%CI = 1.40–3.07). Self-rated average and better school performance are the most significant protective factors against self-injury. While studying at a vocational school or ordinary school in comparison to a key school (AOR = 3.58 and 2.00, respectively; 95%CI = 1.25–10.27 and 1.77–2.26, respectively), being a female (AOR = 1.61, 95%CI = 1.12–2.32), smoking during the past 30 days (AOR = 2.03, 95%CI = 1.52–2.72), having bully experiences, and having mental problems (AOR ranged from 1.86 to 3.73) would increase the odds of conducting self-injury. The sex-stratified model showed similar results (see Appendix A).

## 4. Discussion

This study described the prevalence rate of self-injury and sleep insufficiency among middle school students of different school types in a district of Shanghai, China. The study further examined the effect of sleep insufficiency on self-injury behaviors. The result showed that about 8.42% of middle school students had self-injury, with girls more than boys and ordinary school students more than key school students. Sleep insufficiency is widespread, with approximately four-fifths of students not having achieved the recommended hours for sleep duration. Sleep insufficiency was significantly associated with self-injury behaviors in the statistics, especially among ordinary and vocational school students, though insufficient sleep is most prevalent among key school students. Such association still exists even when we dichotomized the sleep length using the median value (see Appendix A). Other risk factors related to self-injury include being female, worse-off school performance, smoking during the past 30 days, being moderately or severely bullied, and having moderate or severe mental health problems. 

The rate of self-injury reported here is similar to studies conducted among Australian and Norwegian adolescents of a similar age [6,23]; however, it is less than that among Hongkong (23.5%) and US samples (18%), which are approximately around 20% [24,25]. Self-injury is usually considered to be a maladaptive coping strategy to regulate emotion: such as managing or distracting from the negative or unwanted feelings, expressing feelings that feel overwhelming and hard to share with others, feeling a sense of control, or punishing themselves for things they believe they have done wrong [1]. In our study, perceived serious or moderate mental problems accounted for some association between sleep duration and self-injury. Still, the latter association remained significant even in the fully adjusted analyses. The detected correlated factors of self-injury in our study reproduced the results of other research [2,5,19,26]. We found that school type and poor academic performance were significantly correlated with self-injury, echoing a meta-analysis result that suicide and self-injury were higher in those with more academic pressure [24]. When adjusted for all the possible factors, being at a key school other than at an ordinary or vocational school was a protective factor for conducting self-injury. It might be because students in key schools have a higher self-efficacy and self-esteem than their counterparts in ordinary and vocational schools [27]. The “healthy worker effect” could also explain such a phenomenon: the “cleverest ones” being selected by key schools are usually those who are mindful, saying, having a high attention to and awareness of what is occurring in the present; thus, few of them would use self-injury as an improper way to regulate this emotion [18].

Bully victimization experiences and mental problems cooccurred with both self-injury and sleep insufficiency, indicating that negative emotionality could be a plausible pathway to both. Such a relationship suggested that school life-related stressors, for instance, academic demand, social acceptance, etc., are likely to be the predominant ones for adolescents to overcome. Therefore, future interventions should consider adolescents’ cognitive and emotional conditions, and reduce their maladaptive attribution to themselves [25].

Theoretically, the emotional regulating mechanism can serve as a helpful framework for understanding the relationship between sleep and self-injury [28]. Brain imaging studies showed that, with the structural and functional changes of the brain in adolescence, especially that to the prefrontal cortex (PFC), adolescents and young adults are especially prone to a higher stress responsivity and emotional reactivity [29]. Insufficient sleep is probably more likely to happen among those who could not keep the normative sleep duration as their peers to satisfy the time spent on sports/physical training, academic demand, and necessary social networks, which affected their PFC development [30]. In addition, those with internal problems caused by relationship or social acceptance issues would have lower impulsive control inhibition and difficulties in sleep acquisition [6]. What is more, from the transferable view of the studies on suicide, self-injury could also be related to subjective misconceptions, such as being a burden to the family, blaming oneself to be stupid, having no one to care, and perceiving the quality of social relationship as dissatisfactory, which would also affect their sleep quality [12].

Some weaknesses should be kept in mind when interpreting the results. First, the cross-sectional design prohibited the causal interpretation between insufficient sleep and self-injury. However, the association between the two is still informative for future study and intervention design. Second, this study only assessed sleep duration by dichotomizing it into sufficient and insufficient. Previously, studies have reported a U-shape relationship between sleep duration and self-injury [31]. Future studies with more accurate sleep-length assessments are needed to verify if the nonlinear relationship exists among all school-type students. Third, the questionnaire did not distinguish self-injury with suicidal intent from non-suicidal self-injury. However, given that most self-injury, especially in teens and young adults, is non-suicidal, our result is still comparable to studies in this area. Fourth, though we did not include the frequency of self-injury in our research due to the small percentage of occurrence, as self-injury is often cyclical, it is an important point to inspect, which should be taken into consideration in future studies. Lastly, the findings are subject to response biases because of the self-reported questionnaire. Self-injury is a sensitive question and adolescents may tend to under-report it. However, self-report measures remain a valuable source of information in epidemiologic studies. Longitudinal research is needed in the future to verify the relationship, and it would also be necessary to determine whether other changeable factors mediate this relationship.

## 5. Conclusions

Given that less than one in four adolescents had the recommended hours of sleep and the adverse effects of sleep insufficiency on physical and mental health are largely documented, such a phenomenon should raise more public health concerns. The finding of this study strongly suggested that policies that seek to avoid the behavioral expression of the negative affective experience of adolescents, such as self-injury and suicide attempt, and those that attempt to enhance adolescent sleep may be more closely intertwined than previously assumed. Sleep insufficiency and self-injury may possibly be reduced by national government policies regarding school types and educational components. Interventions to work on reducing the academic demand and school bullying, as well as transforming the social norms on praising the good and blaming the bad students, may also play a role in the prevention. In all, a multifaceted approach considering the roles of both school environment and social contexts are needed for adolescents’ well development.

## Figures and Tables

**Table 1 ijerph-19-15595-t001:** Demographic characteristics, and study variables of respondents by sex and by school type.

		Total	Sex		χ^2^/*t* Test	*p*	School Type			χ^2^/F Test	*p*
			Male N = 534	Female N = 558			Ordinary School N = 241	Key School N = 407	Vocational School N = 444		
**Demographics**											
Sex	Male	48.90	-	-	-	-	58.92	48.89	43.47	14.928	0.001
	Female	51.10	-	-			41.08	51.11	56.53		
School type	Ordinary school	22.07	26.59	17.74	14.928	0.001	-	-	-	-	-
	Key school	37.27	37.27	37.28			-	-	-		
	Vocational school	40.66	36.14	44.98			-	-	-		
Age		16.17 ± 1.82	16.03 ± 1.92	16.30 ± 1.71	2.435	0.015	13.36 ± 1.29	16.85 ± 0.92	17.08 ± 0.97	1157.71	<0.001
School Performance	Worse off	26.97	30.25	23.86	6.373	0.041	20.92	27.39	29.93	19.552	0.001
	Average	34.18	31.41	36.79			28.87	34.17	37.12		
	Better off	38.86	38.34	39.34			50.21	38.44	32.95		
Ethnicity	Han	98.35	97.94	98.74	1.084	0.298	96.23	98.28	99.55	10.508	0.005
	Minority	1.65	2.06	1.26			3.77	1.72	0.45		
Family structure	Both parents	81.74	81.61	81.87	0.012	0.914	78.33	87.93	77.93	16.617	<0.001
	Non or one parent	18.26	18.39	18.13			21.67	12.07	22.07		
**Variables of interest**											
Self-injury	Yes	8.42	7.68	9.14	0.756	0.385	12.03	6.14	8.56	6.825	0.033
	No	91.58	92.32	90.86			87.97	93.86	91.44		
Sleep	Insufficient	81.87	79.78	83.87	3.084	0.079	76.35	91.89	75.68	43.964	<0.001
	Sufficient	18.13	20.22	16.13			23.65	8.11	24.32		
**Other covariate variables**											
Self-rated weight	Lighter than average	19.69	28.65	11.11	71.631	<0.001	19.92	16.71	22.30	8.950	0.062
	Average	35.07	37.08	33.15			40.25	35.87	31.53		
	Heavier than average	45.24	34.27	55.73			39.83	47.42	46.17		
Bully victimization	None or Light	60.35	47.75	72.40	75.459	<0.001	55.60	66.09	57.66	13.874	0.008
	Moderate	20.88	25.09	16.85			19.92	19.90	22.30		
	Heavy	18.77	27.15	10.75			24.48	14.00	20.05		
Mental health Problems	None or Light	38.64	61.61	47.85	20.807	<0.001	56.02	29.98	37.16	69.156	<0.001
	Moderate	40.84	26.40	32.44			29.88	53.07	35.59		
	Heavy	20.51	11.99	19.71			14.11	16.95	27.25		
Physical activity	Active	45.60	57.30	34.41	57.657	<0.001	73.86	38.33	36.94	99.687	<0.001
	Non-active	54.40	42.70	65.59			26.14	61.67	63.06		
Smoking during the past 30 days	Yes	3.85	5.24	2.51	5.517	0.019	0.41	0.98	8.33	40.869	<0.001
	No	96.15	94.76	97.49			99.29	99.02	91.67		
Drinking during the past 30 days	Yes	26.56	28.46	24.73	1.950	0.163	14.52	22.85	36.49	43.207	<0.001
	No	73.44	71.54	75.27			85.48	77.15	63.51		
Time spent on video games	Over 4 h	21.61	20.79	22.40	0.420	0.517	11.62	13.27	34.68	75.724	<0.001
	Less than 4 h	78.39	79.21	77.60			88.38	86.73	65.32		

**Table 2 ijerph-19-15595-t002:** Bivariate association of study variable and self-injury.

		No (%)	Yes (%)	Crude OR	95%CI	*p*
Age		16.22 ± 1.80	15.70 ± 1.98	0.87	0.78–0.97	0.010
Sex	Male	49.30	44.57	1		
	Female	50.70	55.43	1.21	0.79–1.86	0.385
Ethnicity	Han	98.50	96.74	1		
	Minority	1.50	3.26	2.21	0.63–7.77	0.218
Living with both parents	Yes	82.06	78.26	0.79	0.47–1.32	0.367
	No	17.94	21.74	1		
School type	Key	93.86	6.14	1		
	Ordinary	87.97	12.03	2.09	1.19–3.66	0.010
	Vocational	91.44	8.56	1.43	0.85–2.41	0.181
School performance	Worse off	25.84	39.33	1		
	Average	35.24	22.47	0.42	0.24–0.74	0.003
	Better off	38.92	38.20	0.65	0.39–1.06	0.084
Self-rated weight	Lower than average	19.60	20.65	1.23	0.67–2.26	0.506
	Average	35.50	30.43	1		
	Higher than average	44.90	48.91	1.27	0.78–2.08	0.340
Physical active	Yes	45.50	46.74	1.05	0.69–1.61	0.819
	No	54.50	53.26	1		
Smoking	Yes	3.40	8.70	2.71	1.21–6.03	0.015
	No	96.60	91.30			
Drinking	Yes	25.70	35.87	1.62	1.03–2.53	0.036
	No	74.30	64.13	1		
Playing video games	Less than 4 h	78.60	76.09	1		
	Over 4 h	21.40	23.91	1.15	0.70–1.91	0.575
Bully Experience	None	62.70	34.78	1		
	Moderate	20.30	27.17	2.41	1.40–4.17	0.002
	Heavy	17.00	38.04	4.03	2.43–6.71	<0.001
Mental Problem	Light	40.40	19.57	1		
	Moderate	41.20	36.96	1.85	1.03–3.33	0.040
	Heavy	18.40	43.48	4.88	2.72–8.74	<0.001
Sleep	Insufficiency	81.70	83.70	1.15	0.65–2.05	0.635
	Sufficiency	18.30	16.30	1		

**Table 3 ijerph-19-15595-t003:** Bivariate association of study variables and sleep insufficiency.

Varialbes	Crude OR	95%CI	*p*
School type			
Ordinary schools vs. key schools	0.28	0.18–0.45	<0.001
Vocational schools vs. key schools	0.27	0.18–0.42	<0.001
Age	0.97	0.89–1.06	0.468
Sex			
Female vs. male	1.32	0.97–1.80	0.080
Ethnicity			
Minority vs. Han	0.57	0.20–1.62	0.293
Living with both parents			
Yes vs. no	0.73	0.48–1.12	0.147
School performance			
Average vs. worse off	1.19	0.79–1.80	0.396
Better off vs. worse off	0.89	0.61–1.30	0.545
Self-rated weight			
Lower than average vs. average	0.97	0.64–1.46	0.884
Higher than average vs. average	1.55	1.09–2.20	0.015
Physically active			
Yes	0.75	0.55–1.02	0.065
Smoking			
Yes	0.48	0.24–0.94	0.031
Drinking			
Yes	0.87	0.62–1.23	0.432
Playing video games			
Over 4 h	0.86	0.60–1.24	0.422
Bully experience			
Moderate vs. none	1.21	0.80–1.83	0.377
Severe vs. none	0.67	0.46–0.98	0.040
Mental problem			
Moderate vs. light	1.95	1.37–2.77	<0.001
Severe vs. light	1.39	0.92–2.10	0.114

**Table 4 ijerph-19-15595-t004:** Correlation of self-injury and sleep insufficiency among middle school students using logistic regression with the standard error of school type clustered.

	Bivariate			Multivariate		
	OR	95%CI	*p*	AOR	95%CI	*p*
Sleep insufficiency	1.15	1.06–1.25	0.001	2.08	1.40–3.07	<0.001
School type (key school)						
Ordinary school	2.09	-	-	3.58	1.25–10.27	0.018
Vocational school	1.43	-	-	2.00	1.77–2.26	<0.001
Interactions						
Ordinary school × Sleep insufficiency	-	-	-	0.53	0.38–0.75	<0.001
Vocational school × Sleep insufficiency	-	-	-	0.59	0.48–0.74	<0.001
Age	0.87	0.85–0.89	<0.001	0.94	0.70–1.26	0.679
Sex						
Female	1.21	0.86–1.69	0.268	1.61	1.12–2.32	0.011
Ethnicity (Han)						
Minority	2.21	1.41–3.46	0.001	1.94	1.00–3.79	0.052
Living with both parents						
Yes	0.79	0.61–1.00	0.050	0.89	0.63–1.25	0.493
School performance						
Average vs. worse off	0.42	0.37–0.48	<0.001	0.43	0.34–0.55	<0.001
Better off vs. worse off	0.65	0.49–0.85	0.002	0.65	0.45–0.94	0.021
Self-rated weight (average)						
Lower than average	1.23	0.72–2.09	0.448	1.18	0.70–1.98	0.531
Higher than average	1.27	0.98–1.65	0.075	1.14	0.87–1.49	0.352
Smoking						
Yes	2.71	1.62–4.51	<0.001	2.03	1.52–2.72	<0.001
Drinking						
Yes	1.62	1.39–1.88	<0.001	1.28	0.92–1.79	0.149
Physically active						
Yes	1.05	0.53–2.08	0.887	1.12	0.61–2.07	0.720
Playing video game						
Over 4 h	1.15	0.67–2.00	0.610	0.86	0.61–1.23	0.410
Bully experience						
Moderate vs. none	2.41	1.75–3.33	<0.001	2.20	1.50–3.23	<0.001
Severe vs. none	4.03	2.06–7.91	<0.001	2.82	1.07–7.48	0.037
Mental problem						
Moderate vs. light	1.85	1.18–2.90	0.007	1.86	1.05–3.30	0.035
Severe vs. light	4.88	3.07–7.75	<0.001	3.73	2.21–6.30	<0.001

Note: All 95%CIs are calculated on the school-type clustered variances. Multivariate logistic regression model included the interaction between sleep insufficiency and school type.

## Data Availability

Data are available upon reasonable request. Please contact the corresponding authors for detail.

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
