# Peer review of "The Link between Sleep Insufficiency and Self-Injury among In-School Adolescents: Findings from a Cross-Sectional Survey of Multi-Type Schools in Huangpu District of Shanghai, China"

_ijerph, 2022, doi:10.3390/ijerph192315595_

Round 1
Reviewer 1 Report
The authors have addressed a very interesting and important topic.
Overall, I would recommend copy editing for clarity throughout.
For the international reader, it would be helpful to include a description/definition of the different school types – I am not familiar with key schools, for example. The introduction should provide some background on why we might expect differences between these school types. In the discussion, I would like to see some explanation/analysis of why these differences were found and what the implications of these findings might be.
Though analyses adjusted for sex, I am wondering if you considered stratifying regressions by sex (did you examine interactions with sex?).
One shocking result of the study was the huge majority of adolescents with insufficient sleep! I would love to see this discussed in more detail. I would also consider a sensitivity analysis using a different cut-off (perhaps the median amount of sleep, or lowest quartile?), since less sleep than recommended seems to be the norm among this cohort.
The discussion should be edited to soften causal language (e.g., sleep insufficiency “worsens” self-injury). Discussion of mechanisms (academic demands, lower impulse control, cognitive distortions) is highly speculative and in need of references. You briefly mention other plausible pathways, e.g., negative emotionality leads to both poorer sleep and self-injury, but the implications of this should be discussed in more detail.
Author Response
Overall, I would recommend copy editing for clarity throughout.
Response: Thank you very much. The authors read and organized the paper carefully this time. In addition, we used online editing software to detect and revise grammar errors.
For the international reader, it would be helpful to include a description/definition of the different school types – I am not familiar with key schools, for example. The introduction should provide some background on why we might expect differences between these school types. In the discussion, I would like to see some explanation/analysis of why these differences were found and what the implications of these findings might be.
Response: We rechecked the official files that define the school types in Huangpu district, and found we had made a mistake previously: we confused the names of key schools with ordinary schools and corrected them in this revision. We are so grateful for your reminder and suggestions. Otherwise, we would make a huge mistake.
As follows, the description and implication of different school types were added in the introduction and discussion parts, respectively:
INTRODUCTION
“Specifically, in the current Chinese context, schooling is becoming the most important investment for the family. After the 9-year compulsory schooling, about half of the junior high students who wanted to pursue higher education would go to senior high school, with the “cleverest” ones being preferred by key senior high schools and the ordinary students going to the ordinary senior high school; while the other half would go to the vocational schools with less academic pressure. With tremendous academic pressure, Chinese students who wanted to get better performance are even forced to sacrifice sleep time.” (line 61-68, page 2)
DISCUSSION
When adjusted for all the possible factors, being at a key school other than an ordinary or vocational school was a protective factor for conducting self-injury. It might be because students in key schools have higher self-efficacy and self-esteem than their counterparts in ordinary and vocational schools. The “healthy worker effect” could also explain such a phenomenon: The “cleverest ones” being selected by key schools are usually those who are mindful, saying having high attention to and awareness of what is occurring in the present, and thus few of them would use self-injury as an improper way to regulate this emotion. (line 237-244, page 8)
Sleep insufficiency and self-injury may possibly be reduced by national government policies regarding school types and educational components. (line 291-292, page 9)
Though analyses adjusted for sex, I am wondering if you considered stratifying regressions by sex (did you examine interactions with sex?).
Response: Thank you for asking this question. The bivariate analysis did not find any significant association between self-injury and sex(P=0.385). Thus, we decided to enter “sex” as a control variable into the model.
In the data exploration, we did try the stratified model. We have added a table in the supplementary file using the sex-stratified model. However, in the sex-stratified model, the problem of ordinary school is underestimated. Thus, we thought the model we used in the main text is better.
In the multivariate model, the interactions of sleep and sex were also examined, just like what we did with school types. However, there was no significant interaction between sleep and sex, thus, we did not include the interactions.
One shocking result of the study was the huge majority of adolescents with insufficient sleep! I would love to see this discussed in more detail. I would also consider a sensitivity analysis using a different cut-off (perhaps the median amount of sleep or lowest quartile?), since less sleep than recommended seems to be the norm among this cohort.
Response: Thank you again for raising this question out. The sleep length options for our adolescents included "less than 7h", "7-8h", "8-9h", and "over 9 h". Because the participants in our study include both middle school students (from grade 6-9) and high school students (from grade 10-12), we thought recoding the sleep length variable into two groups with the recommended time duration according to adolescents’ age would be proper. Besides, it would be helpful for the readers and policymakers to know what the real situation of adolescents’ sleep was (And you are right, the result is shocking, which is the part we wanted to show).
In the data exploration process, we did use “less than/ over 7h” as a cut-off point based on the median sleep duration the adolescents reported. The results are included in supplement table 2, similar to the results we presented in the main text, for your and the readers’ reference. As such, we added in the discussion that “Such association still exists even we dichotomized the sleep length using the median value (see supplementary table 2).”(line 219-221, page 8)
Because the question only asked “over 9 h” to the right censor, and the number in this group is too small (only 52 students), we find it hard to group the options into three categories. Otherwise, the multivariate logistic regression model would be unstable.
We have addressed in limitations as follows:
“Second, this study only assessed the sleep duration by dichotomizing it into sufficient and insufficient. Previously studies have reported a U shape relationship between sleep duration and self-injury[31]. Future studies with more accurate sleep length assessment are needed to verify if the nonlinear relationship exists among all school-type students.” (line 269-273, page 9)
The discussion should be edited to soften causal language (e.g., sleep insufficiency “worsens” self-injury). Discussion of mechanisms (academic demands, lower impulse control, cognitive distortions) is highly speculative and in need of references. You briefly mention other plausible pathways, e.g., negative emotionality leads to both poorer sleep and self-injury, but the implications of this should be discussed in more detail.
Response: Thank you for your reminder and for the insights. The discussion was revised accordingly.
Revision of causal language:
Sleep insufficiency worsens were significantly associated with self-injury behaviors in statistic…” (line 218-219, page 8)
The reference related to the mechanisms are added as follows:
[28 ]Khazaie H, Zakiei A, McCall WV, et al. Relationship between Sleep Problems and Self-Injury: A Systematic Review. Behav Sleep Med 2021;19:689-704.
[29] Guyer AE, Silk JS, Nelson EE. The neurobiology of the emotional adolescent: From the inside out. Neurosci Biobehav Rev 2016;70:74-85.
[30] Dimitrov A, Nowak J, Ligdorf A, et al. Natural sleep loss is associated with lower mPFC activity during negative distracter processing. Cogn Affect Behav Neurosci 2021;21:242-253.
[6] Hysing M, Sivertsen B, Stormark KM, et al. Sleep problems and self-harm in adolescence. Br J Psychiatry 2015;207:306-312.
[12] Baiden P, Tadeo SK, Tonui BC, et al. Association between insufficient sleep and suicidal ideation among adolescents. Psychiatry Res 2020;287:112579.
Plausible pathways regarding negative emotionality were discussed in more detail as follows:
Bully victimization experiences and mental problems cooccurred with both self-injury and sleep insufficiency, indicating that negative emotionality could be a plausible pathway to both. Such a relationship suggested that school life-related stressors, for instance, academic demand, social acceptance, etc., are likely to be the predominant ones for adolescents to overcome. Therefore, future interventions should consider adolescents’ cognitive and emotional conditions and reduce their maladaptive attribution to themselves. (line 244-250, page 9)
Reviewer 2 Report
The study describes the prevalence rate of self-injury and sleep insufficiency among 203 middle school students of different school type in a district of Shanghai, China, as well as the effect of sleep insufficiency on self-injury behaviors.
I wonder, why did you decide to use 0/1 coding (and not continuous) for self-injury and some covariates?
Overall, I think the issue of sleep insufficiency in teenagers is a serious problem and may possibly be reduced by national governments policies regarding school structure and educational components.
Author Response
I wonder, why did you decide to use 0/1 coding (and not continuous) for self-injury and some covariates?
Response: Thank you for raising this interesting question.
Generally, by dichotomizing, we're asserting that there is a straight line of effect between one variable and another, such as conducting or not conducting self-injury, living with both parents or not, smoking or not, sufficient or insufficient sleep, etc. And by doing so, the results of statistical analysis are straighter to explain and are easier for readers and policymakers to digest.
However, dichotomization may also increase the risk of a positive result being a false positive. And may underestimate the extent of variation in outcome between groups (e.g., in our result, considerable variability between one-time self-injury and multiple-time self-injury may be subsumed within each group, which we've addressed in the limitation part).
Individuals near the cut-off point are characterized as being very different rather than very similar. To be cautious about such problems, we've conducted a sensitive analysis using another cut-off point to check if the main result was consistent, and the answer is yes.
Thirdly, using two groups conceals any nonlinearity in the relation between the variable and outcome. In our case, we cannot make sure if the effect of sleep length on self-injury would be nonlinearity as we do not have sufficient information. We've addressed this in the limitation part and hope we can collect more detailed information using more objective devices like wearable smart bands in the future.
Ref: Altman DG, Royston P. The cost of dichotomising continuous variables. BMJ. 2006 May 6;332(7549):1080. doi: 10.1136/bmj.332.7549.1080.
Overall, I think the issue of sleep insufficiency in teenagers is a serious problem and may possibly be reduced by national governments policies regarding school structure and educational components.
Response: Thank you very much for the comments. That is also what we thought, and we are grateful to your insights.